# Addition of Phosphogypsum to Fire-Resistant Plaster Panels: A Physic–Mechanical Investigation

**Abdessalam Guedri** [1,2,*]**, Fatma Abdallah** [3]**, Nourhen Mefteh** [3]**, Noureddine Hamdi** [4,5]**, Oscar Baeza-Urrea** [2]**, Jean-Frank Wagner** [2,*] and **Mohamed Faouzi Zagrarni** [5]

1    Department of Geology, Faculty of Sciences of Bizerte, Carthage University, Bizerte 7021, Tunisia
2    Department of Geology, Trier University, 54296 Trier, Germany
3    Department of Geology, Faculty of Sciences of Gabès, Gabès University, Gabès 6072, Tunisia
4    Laboratory of Composite Materials and Clay Minerals, National Center for Research in Materials Sciences (CNRSM), Soliman 8020, Tunisia
5    Higher Institute of Water Science and Techniques of Gabes, Gabès University, Gabès 6072, Tunisia
*    Correspondence: guedri.abdesslem@fsb.u-carthage.tn (A.G.); wagnerf@uni-trier.de (J.-F.W.)

**Abstract:** Gypsum (GPS) has great potential for structural fire protection and is increasingly used in construction due to its high-water retention and purity. However, many researchers aim to improve its physical and mechanical properties by adding other organic or inorganic materials such as fibers, recycled GPS, and waste residues. This study used a novel method to add non-natural GPS from factory waste (phosphogypsum (PG)) as a secondary material for GPS. This paper proposes to mix these two materials to properly study the effect of PG on the physico-mechanical properties and fire performance of two Tunisian GPSs (GPS1 and GPS2). PG initially replaced GPS at 10, 20, 30, 40, and 50% weight percentage (mixing plan A). The PGs were then washed with distilled water several times. Two more mixing plans were run when the pH of the PG was equal to 2.4 (mixing plan B), and the pH was equal to 5 (mixing plan C). Finally, a comparative study was conducted on the compressive strength, flexural strength, density, water retention, and mass loss levels after 90 days of drying, before/after incineration of samples at 15, 30, 45, and 60 min. The results show that the mixture of GPS1 and 30% PG (mixing plan B) obtained the highest compressive strength (41.31%) and flexural strength (35.03%) compared to the reference sample. The addition of 10% PG to GPS1 (mixing plan A) improved fire resistance (33.33%) and the mass loss (17.10%) of the samples exposed to flame for 60 min compared to GPS2. Therefore, PG can be considered an excellent insulating material, which can increase physico-mechanical properties and fire resistance time of plaster under certain conditions.

**Keywords:** gypsum plaster; phosphogypsum; physico-mechanical properties; fire performance

## 1. Introduction

Gypsum is a natural calcium sulfate hydrate (GPS, $CaSO_4.2H_2O$), converted to calcium sulfate hemihydrate (P, $CaSO_4.1/2H_2O$) when heated. The hemihydrates form is known for its low bulk density and relatively good ductility. The importance of this material is reflected in Thailand (9.3 million tons (MT)), the United States (20 MT), Spain (7 MT), Iran (16 MT), and China (16 MT), according to the 2019 statistics investigation [1]. In Tunisia, the "Company of Phosphate Gafsa (CPG)" reported that Tunisia had an annual GPS extraction capacity of 3.73 MT in 2019 [2]. However, Tunisia's annual GPS production remains modest compared to world production, although it ranks fourth in GPS availability. It may be for economic reasons. Since the Jurassic (Toarcian–Bajocian) geological epoch, huge GPS volumes are mainly present in the thick blocks of the Mestaoua geological formation in the Upper Lias epoch (Figure 1). The National Minerals Office estimates the total volume of Mestaoua's GPS to be 10 to 15 km in length and 400 to 600 m deep [3]. This plaster is used in various fields such as construction (decoration, wall, and ceiling mortar, etc.), medical

applications, etc. Especially in the area of building construction, current scientific trends are moving towards the invention of light green materials with excellent thermomechanical and acoustic properties [4–6]. These properties' performance depends on the material used in the composite design. Often, during sample preparation, the by-products are used, such as sand (dunes, rivers, etc.), gravel, organic and industrial waste (phosphogypsum (PG)), fibers (lignocellulose, glass, plastic, among others), etc. [7–11].

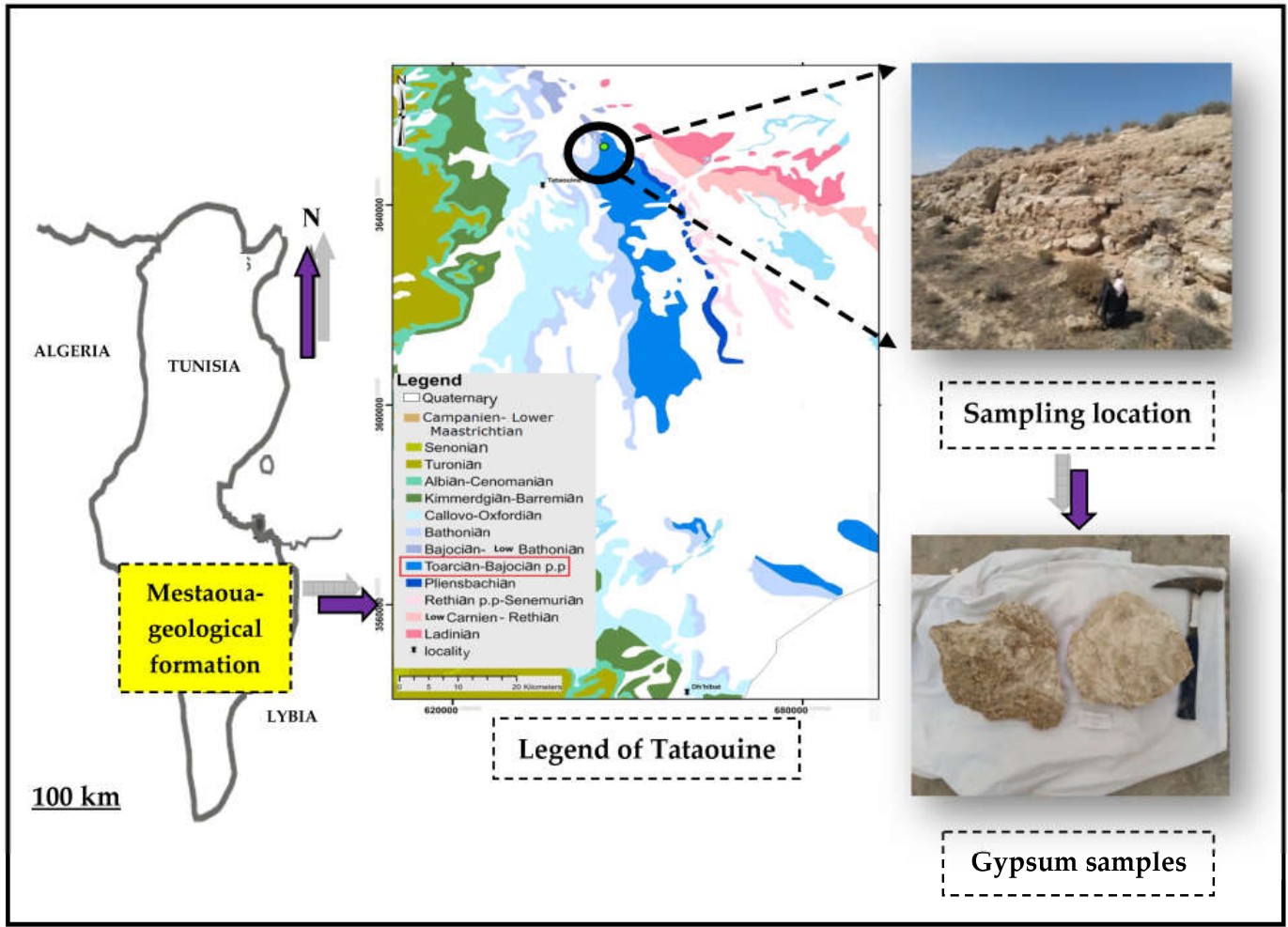

**Figure 1.** The geographic, geological location (extract from the geological map 1/500,000) of the gypsum deposit in south Tunisia and their sampling area.

On the other hand, PG is a material similar to GPS except that it comes from industrial wastewater and contains polluting impurities, which are estimated to be up to 258 MT in 2018 [12]. The chemical formula of PG is as follows:

$$Ca_{10}(P0_4)_6F_2 + 10H_2SO_4 + 20\,H_2O \rightarrow 6H_3PO_4 + 10(CaSO_4 + 2H_2O) + 2HF \qquad (1)$$

The amount of this secondary raw material produced varies according to each country's national wealth of phosphates, which is considered the raw material to obtain the PG. In Tunisia, commercial phosphate production exceeded the 2 MT threshold during the first half of 2022 [13]. Phosphate production has put Tunisia among the top five phosphate producers in the world. More than 80% of phosphate production in Tunisia is converted into commercial products (acid $H_3PO_4$, $H_2SO_4$, etc.), superphosphate fertilizers, detergents, and food additives). Phosphate conversion was carried out at "Tunisian Chemical Group (GCT)" more than 30 years. In the Sfax region of Tunisia, GCTs accumulate industrial emissions of

PG on a large scale, as in China, Poland, Russia, Morocco, and several other countries [14]. Zairi et al. show that PG impurities in fluorine, phosphate, and heavy metals cause soil and groundwater contamination at Sfax storage sites, exceeding Tunisian waste standards [15]. Additionally, the PG negatively affects fisheries and tourism as this material is radioactive and difficult to biodegrade, causing harmful effects on aquatic plants (decrease in vegetation), marine animals, and even humans [16]. As a result, socio-political issues arise as civil society organizations do not accept the release of this pollutant, and rapid intervention is required to address the issue. To solve the problem of PG, researchers studied the potential uses of PG [17]. They have attempted to reduce heavy metals and degrade soil organic pollutants by using phytoremediation (detoxification of plants, e.g., vascular plants, algae, fungi). Costa et al. showed that soil amendment with a mixture of 75% red mud and 25% PG could reduce arsenic availability in *Urochloabrizantha* planting areas [18]. Using PG as fertilizer also decreased soil salinity and sodicity while increasing soil fertility and soil organic matter [19]. Some other researchers have suggested using PG in the field of civil engineering. PG treatment processes have been developed to clean and neutralize the residual acidity of this secondary raw material. However, the scientific method of chemical decomposition (purification) to make PG assimilable with GPS (thermochemical treatment and flotation [20], recover of lanthanides [21], etc.) is not sufficiently profitable for material manufacturers as the cost of the purification process eliminates any economic benefit to the proposed solution, especially where GPS is abundant. However, PG may be used in some countries to fill the natural GPS gap due to the high costs of supplying it, which leads to a tangible increase in construction prices, as is the case with the European Union [22].

The scientific trend is stabilizing towards using PG as an additive to create materials (composite materials and geopolymer) with well-defined ratios [4,9–11,17]. The researchers recommend neutralizing residual PG acid to avoid swelling and unwanted cracking. This can be done by washing the PG with water or by mixing the PG with crushed limestone aggregate [23]. In general, modified PG has been used in the manufacture of cement, geopolymer binders, cementitious composites, brick, paving blocks, ceramic wall tiles, road works, etc., to reduce the manufacturing cost, reduce the environmental risks and improve physico-mechanical properties of samples [9–11]. Researchers produced high-performance binder pastes and mortars with compressive strength up to 90 MPa using PG [4]. In addition, other researchers produced a new type of plasterboard with good fire resistance using PG as the only raw material without using fibers and paper [24].

Fire resistance testing reflects the resistance of a material in the event of a building fire. A more fire-resistant material, i.e., longer time to save people from the collapse of structural elements (walls, ceilings, etc.). Researchers and companies use a variety of recognized international standards (UK, European, US, ISO, Australian, and Chinese standards) or special tests within "Fire Testing of Building Structures" to determine specific physical and mechanical properties (flexural and compressive strength, loss of mass) of a sample after fire-proof combustion process [25]. Researchers point out that the insulating properties of GPS boards are related to the density and fiber content of the GPS boards. High-density GPS boards have a higher heat-absorbing capacity. Adding fibers to GPS boards can effectively delay the shedding of GPS boards caused by fire [5]. In addition, other researchers have successfully developed the flexural strength of a refractory board made from a mixture of water-based intumescent binder, vermiculite, and perlite compared to commercial plaster [25]. PG has other potentially valuable uses that have yet to be discovered. This could include what we did in this study, where we successfully demonstrated for the first time that PG could also be used with GPS from southern Tunisia (Tataouine (GPS1) and Sidi Bouzid (GPS2)) to improve fire resistance.

In this study, GPS plaster was developed at a low manufacturing cost (no fibers or chemical additives) to maintain good physical and mechanical properties, even in the event of a fire, using PG as additive material, and thereby reducing the environmental risk of PG deposits.

## 2. Results and Discussions

### 2.1. Morphological and Physico-Chemical Properties

Scanning electron microscope (SEM).The GPS crystal particles were plate-like, and needle-like, with smooth surfaces and curved edges (Figure 2a). After GPS dehydration, the voids increased significantly, reflecting the high open porosity of the plaster (Figure 2b).

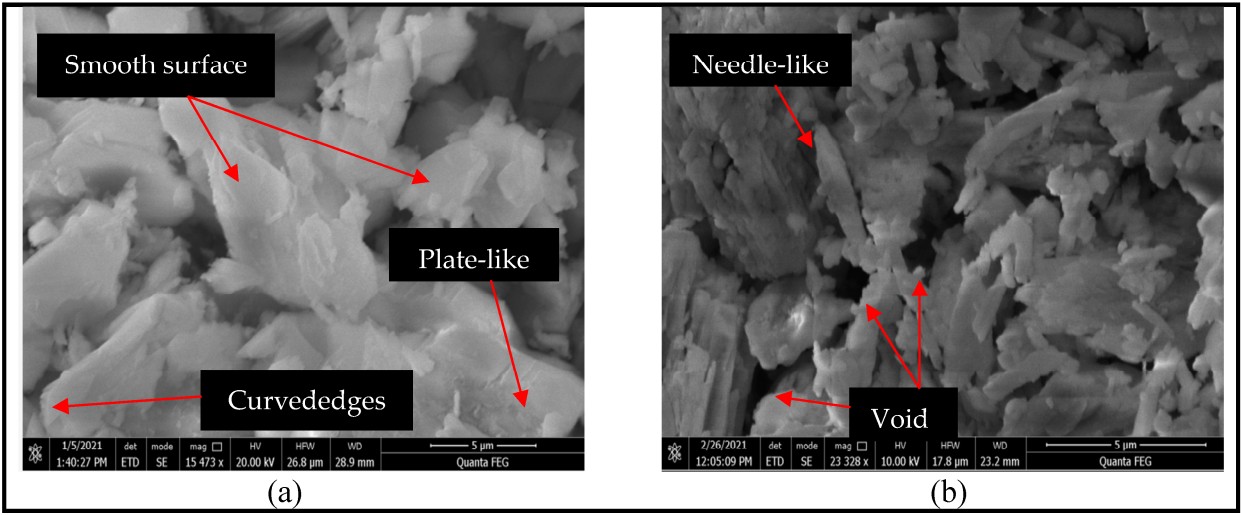

**Figure 2.** SEM images of (**a**) gypsum with magnification ×15,473 and (**b**) plaster with magnification ×23,328.

X-ray diffraction (XRD). The sample GPS contains mainly GPS ($CaSO_4.2H_2O$). The diffractogram shows in addition less calcite ($CaCO_3$) and traces of bassanite ($CaSO_4.1/2H_2O$) (Figure 3).

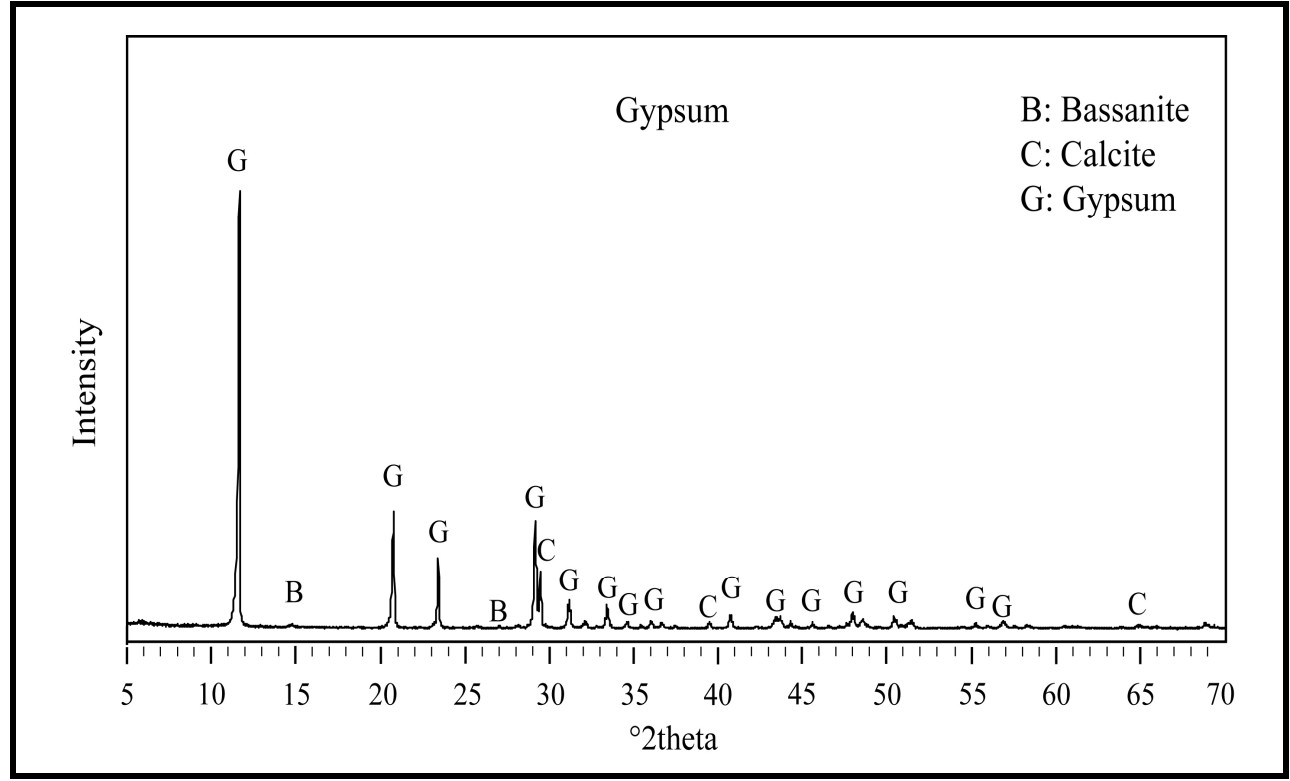

**Figure 3.** XRD pattern of gypsum of the south in Tunisia.

Chemical components. Table 1 illustrates the chemical analysis results of PG after each wash. It is mainly composed of calcium (Ca), phosphate ($PO_4^{-3}$), sulfate ($SO_4^{-2}$), fluorine ($F^-$), magnesium (Mg), traces of metals cadmium (Cd), and iron (Fe). It was observed that the percentage of these chemical elements of PG relative to raw PG decreased with an increasing number of washes (Table 1).

**Table 1.** Chemical analysis of phosphogypsum.

|  | Ca % | $PO_4^{-3}$ % | $SO_4^{-2}$ % | $F^-$ % | Mg % | Cd % | Fe % |
|---|---|---|---|---|---|---|---|
| WS1 * | 96.55 | 22.86 | 74.00 | 1.59 | 32.81 | 91.30 | 7.97 |
| WS2 * | 31.03 | 20.00 | 58.54 | 0.43 | 1.19 | 86.96 | 5.28 |
| WS3 * | 33.45 | 10.29 | 51.22 | 0.38 | 0.37 | 108.70 | 3.68 |
| WS4 * | 33.10 | 9.43 | 52.44 | 0.38 | 0.20 | 69.57 | 3.26 |
| WS5 * | 33.10 | 4.29 | 36.58 | 0.29 | 0.00 | 56.52 | 3.00 |

* WS1-5: Phosphogypsum washed 1 to 5 times. Mg, Cd, Fe and Ca were carried out using an atomic absorption spectrometer (AnalytikjenanovAA 300). $PO_4^{-3}$, $SO_4^{-2}$ and $F^-$ concentration were measured using the ionic chromatography (930-compact-ic-flex).

Acidity and weight loss. Figure 4 shows the pH measurements and percent PG weight loss before and after each wash. It was observed that PG after washing resulted in an increase in pH. The residual acidity of PG decreased from 2.20 (WS0) to 5.00 (WS5). Zairi et al. showed that twelve consecutive washes under the same treatment conditions were required to achieve pH neutrality [10]. However, storage factors for used PG need to be considered, including storage time and climatic influences (e.g., rain may wash away PG in storage), which will certainly affect its chemical concentration and pH. According to Moncer et al., the pH values obtained in the fifth wash (WS5) were obtained in the second wash (pH = 5.17), although the conditions are the same (stirring time and wash water volume) [26]. Therefore, the number of washings of PG cannot be accurately determined because the residual acidity and chemical constituent concentration of PG are related to its storage conditions.

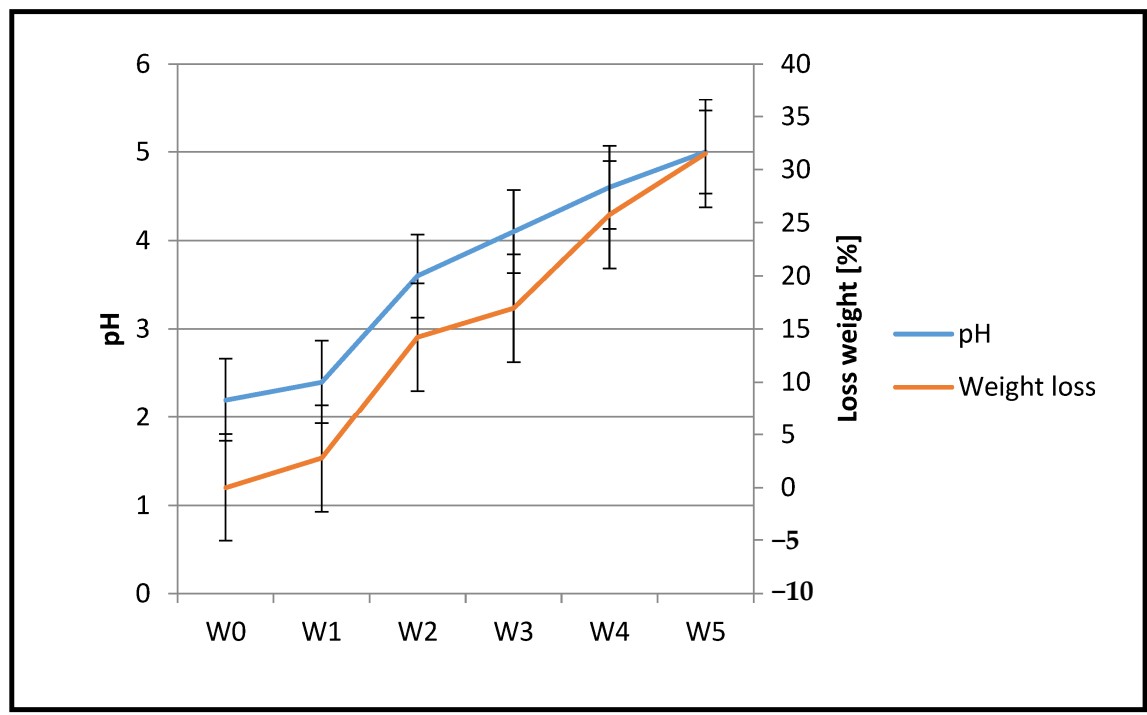

**Figure 4.** Percent weight loss and pH of raw (WS0) and washed (WS1-5) phosphogypsum.

Concerning the loss of PG solids mass after each wash, Figure 4 shows the percent PG weight loss before and after each wash. It was observed that PG after washing resulted in a decrease in the amount of PG. The percent solids weight loss of PG decreased from 2.79% (WS1) to 31.51% (WS5). In addition, attention must be paid to the amount of water used for washing, as the latter contributes to the national and global drinking water crisis. We cannot say we got good results at the WS5 level as we used 20 L of distilled water to bring the pH to 5 and did not exceed 684.82 g of solids (31.51% of loss solid PG). It is not recommended to use potable water for washing PG.

Therefore, it is essential to compare the effect of PG on the physico-mechanical properties of the specimens in the following three cases:

- WS0: 　　no mass loss of PG (0%), with pH = 2.2;
- WS1: 　　low mass loss of PG (2.79%), with pH = 2.4;
- WS5: 　　significant mass loss of PG (31.51%), with pH = 5.

### 2.2. Composites Physical-Mechanical Properties

Effect of phosphogypsum on sample density. Figure 5 shows the density results performed on GPS-based samples. Samples prepared with raw PG and GPS1 or GPS2 showed low density with an increasing amount of PG in the mixture. The density of the plaster powder (1.00 g.cm$^{-3}$) is greater than that of raw PG (0.80 g.cm$^{-3}$) and washed PG (0.86 g.cm$^{-3}$). However, it can be observed that the addition of washed PG to the plaster (GPS1 and GPS2) hurts the density compared to the reference sample (PG0). Compared to the reference sample (1.05 g.cm$^{-3}$), the best sample density was obtained at the PG5A level (0.80 g.cm$^{-3}$) when using 50% raw PG and GPS2. The highest sample density was obtained at the PG4B level (1.24 g.cm$^{-3}$), where 40% "one wash PG" and GPS1. In general, adding raw PG will help reduce the density of the material due to its lower density (0.80 g.cm$^{-3}$) compared to plaster powder (1.00 g.cm$^{-3}$) and by increasing the void volume of the sample. Conversely, the addition of washed PG helped to increase the density of the material, which could be explained by washing to reduce fines and thus improve the particle size distribution of the mixture. Finally, the density results confirmed the importance of raw PG in reducing sample weight compared to washed PG.

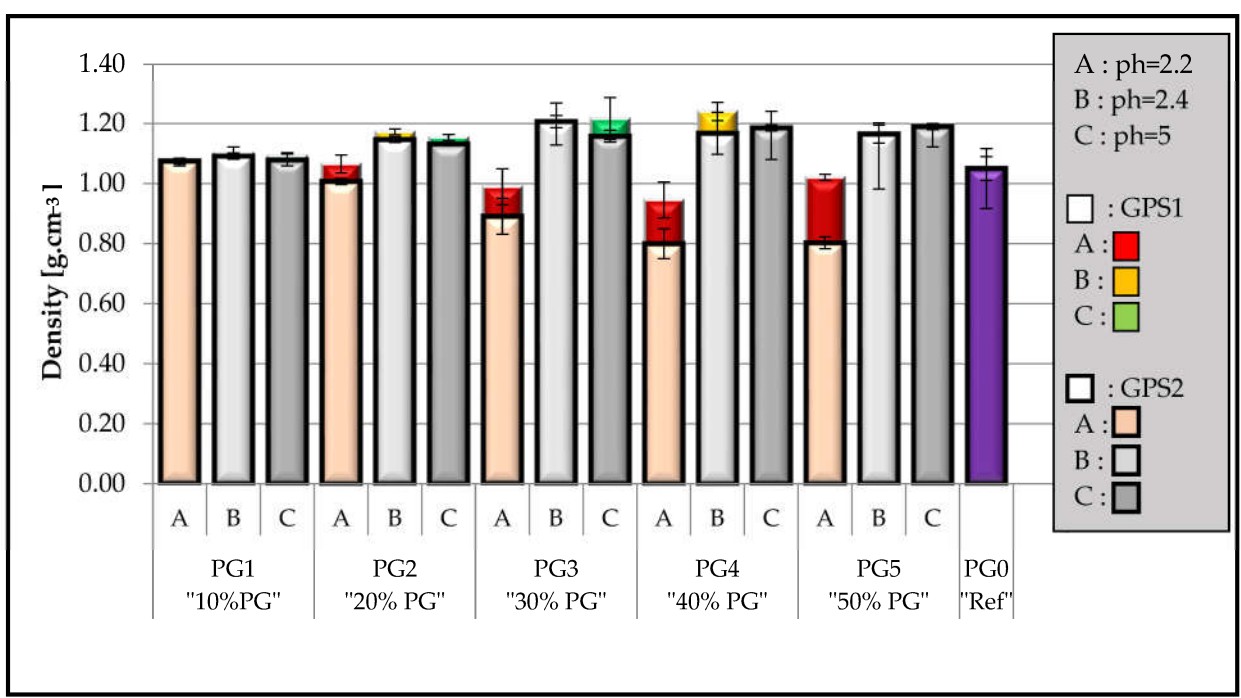

**Figure 5.** Variation in density of plaster under different plans.

Effect of phosphogypsum on sample flexural strength. Figure 6 shows the results of a flexural strength test performed on a GPS-based sample. Samples made with raw PG and GPS2 showed low flexural strength. It can be observed that the mixture of GPS1 with washed PG leads to an increase in flexural strength, especially in PG2B (7.94 MPa) and PG3C (8.62 MPa). Impurities in PG hurt flexural strength. The chemical analysis in Table 1 confirms this observation. In general, the number of PG washes, their mass percentage in the mixture and the purity index of the GPS are essential factors that lead to an increase in the durability of the material compared to the reference samples (PG0). The highest flexural strength was obtained at the PG3C level compared to PG0 (5.60 MPa) when using a 30% "PG wash five times" and GPS1. Finally, the results of the flexural strength test confirmed the importance of PG washing and being used in small quantities before the production of eco-materials, such that GPS has a high purity index like Tataouine GPS (>94%), as shown in Table 2 [8].

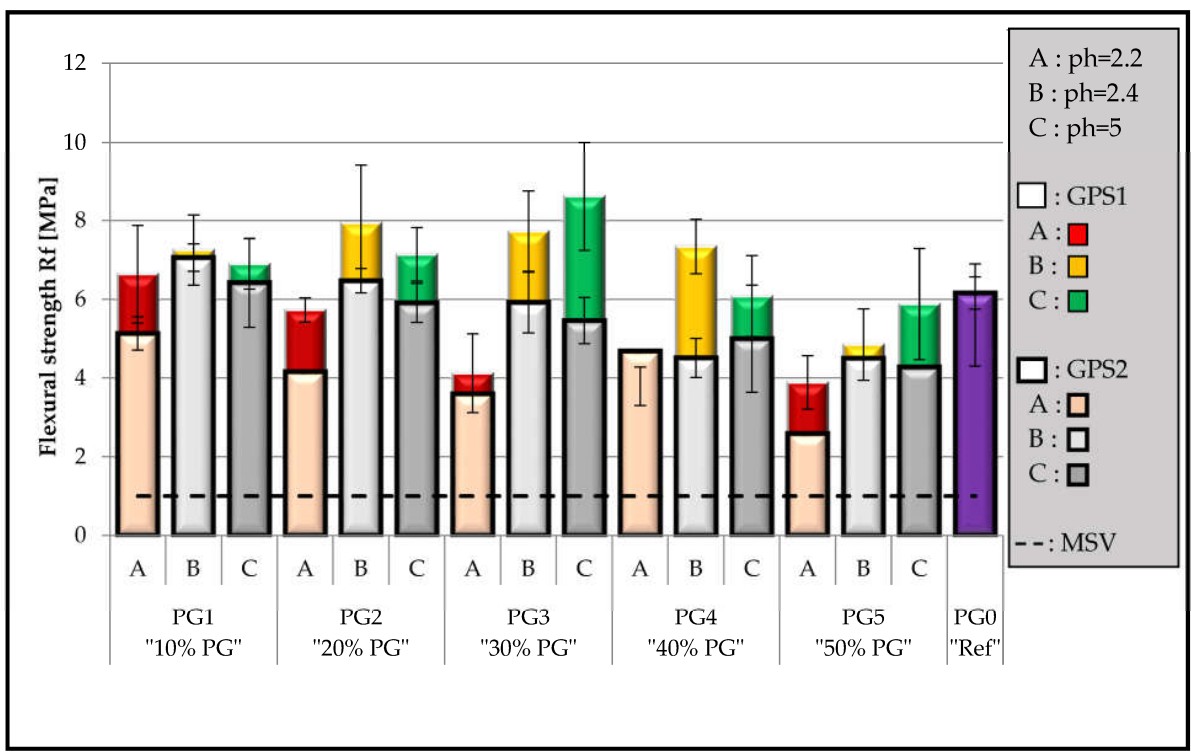

**Figure 6.** Variation in flexural strength of plaster under different plans.

Effect of phosphogypsum on sample density loss after combustion. Figure 7 shows the test results of density loss under flame (15, 30, 45, and 60 min) for material samples (PG0-5) under different conditions of PG (A, B, and C). In fact, as the burning time of the samples increases, their weight loss also increased, so most of the samples cracked before the end of the experiment, while the samples PG1A, PG1B, PG1C, and PG3C that remained refractory after 60 min became brittle (Figures 7 and 8). The results had shown that when the amount of raw PG in the sample was increased, the best low-density loss results were obtained under the flame, as shown in curve A of the mixing plane (blue rhombus). Admittedly, "PG washed five times" is also effective, especially on GPS2, where it appears on the mixing plane curve C (green triangle). Conversely, using more than 10% of "one wash PG" with GPS is not recommended as it will significantly increase the percent weight loss of the sample under a flame, which appears on mixing plane curve B (orange square). The lowest density loss is obtained after 60 min of combustion at the level of PG3C (10.67%) where 30% of "PG washed five times" and 70% of GPS1 have been mixed. Meanwhile, samples of PG1A (made by GPS1 and which has the best compressive strength after the fire), where

10% of "PG brut" and 90% of GPS have been mixed, have a percent density loss of 15.07%. This means that although the fragmentation rate of PG1A is greater than that of sample PG3C, sample PG1A is still resilient for a more extended period of time compared to sample PG3C. This result showed us the positive aspect of the untreated PG in its high ability to maintain the cohesion of the sample even for more than an hour under fire, despite the loss of its structure by a rate greater than the PG washed with distilled water. In fact, the fire resistance of a sample is related to the materials used in its manufacture. As can be seen from chemical Formulas (1) and (2), the chemical composition of these materials includes a large amount of water. When the sample is exposed to fire, some of the chemical composition water absorbs the fire's heatand turns it into steam. This chemical process is called "calcination". Continued application of thermal energy to the sample over time will cause any remaining water from crystallization to evaporate and thus damaging the sample. The reduction in material volume (loss of density) leads to the formation of cracks that cause the sample to break due to structural integrity. The presence of these cracks on the surface of the sample can facilitate the penetration of hot gases into the sample, which explains the increased percentage of density loss with the burning time of the eco-materials. Finally, the percent density loss results after the material burning process demonstrate the importance of washing (Case C) and feedstock (Case A) PG to reduce the amount of material lost and thus ensure a more durable material under flame in the short term.

**Table 2.** Physical and mechanical properties of the plaster of Southern Tunisia [8].

| Properties | Values | | Test Method |
|---|---|---|---|
| Purity | >94% | | ASTM test purity |
| Density | 1.05 (g.cm$^{-3}$) | | Tests for mechanical and physical properties of aggregates |
| Slump test | $180 \pm 10$ (mm) | | Slump test |
| Beginning of setting | 7–8 (min) | | Setting time test |
| End of setting (Shore 40) | 20–22 (min) | | |
| End of setting (Shore 60) | 22–26 (min) | | |
| Mass heat capacity | 1000 (J.kg$^{-1}$.K$^{-1}$) | | Thermal siffusivity measurement |
| Flexural strength (N/mm$^2$) | 2 h >1.5 | 7 days >3 | Three-point bending test |
| Surface hardness (N/mm$^2$) | 2 h >4 | 7 days >10 | Compression strength test |
| Sieve size | Residue at 200 μm ≤1% | Residue at 100 μm ≤10% | ASTM test sieves |

Effect of phosphogypsum on sample compressive strength. Figure 8 shows the results of compressive strength tests performed on GPS-based samples before/after 60 min of flame exposure. Some samples retained their structure after 60 min under the flame, while the residue had broken down before that, as shown in Figure 7. We can see in the compressive strength results the same negative impact that occurs on the flexural strength when using PG and GPS2, and consequently, the same factors (the number of PG washes, their mass percentage in the mixture, and the purity index of GPS) will be able to increase the compressive strength. Another reason that may affect the material performance results is "PG moisture" especially since the moisture content of virgin PG is lower than that of water-washed PG, and the increase in water absorption is reduced by adding "PG washed many times" to the sample. The compressive strength results of PG1A after burning for 60 min are considered compelling evidence that not only the impurities and residual acidity of PG affect the mechanical properties of the samples but also the water content of PG. In general, this includes the amount of plaster used, the water content, pH, impurities and residual salts, and the purity index of the GPS used to have an impact on the mechanical

properties, both before and after the combustion process. The highest compressive strength was obtained at the PG3B level (10.53 MPa) compared to the reference specimen (6.18 MPa), where 30% "once time washed PG" and the GPS1. The highest flame resistance is achieved at the PG1A level (3.6 MPa), where the percent density loss (15.07%), the raw PG (10%), and the GPS1 (90%). This is the first study to show that raw PG positively affects on the mechanical properties of the samples under fire conditions. If the mass of our raw material PG does not exceed 10%, the sample can withstand more than 60 min in the event of a fire in the building. The reason is that the results obtained at the PG1A level (3.6 MPa) are still higher than the minimum compressive strength values (2 MPa) allowed by the standards (MSV). Finally, the compressive strength test results before and after the combustion process of the materials show the importance of washing PG to improve the resistance of specimens and the importance of raw PG to improve the fire resistance of samples and, consequently, passive fire protection.

In summary, we can say that our samples made with washed PG have better mechanical properties than raw PG (Figures 6 and 8). However, compared to PG washed with distilled water, the latter is effective in fire resistance (Figure 8). This may be due to its strong water retention capacity, as it contains high concentrations of chemical elements capable of forming chemical bonds with water as F$^-$ and oxide (Table 1). As the concentration of these chemical elements decreased, the mass loss of the samples increased, followed by a decrease in fire resistance for the longest possible period (Figures 7 and 8). Except for PG3C (where PG was washed 5 times), the mass loss was significantly reduced, as shown in curve B (Figure 5). This exception can be explained by an improvement in the particle size of the mixture, which contributes to a reduction in the porosity of the GPS (Figure 2), thereby increasing the density of the sample, as shown in Figure 5. In general, the higher the proportion of water-washed PG in GPS, the higher its density, mechanical strength, and weight loss rate, but the fire resistance decreases, which is different from raw PG.

Table 3 shows that Tunisian PG seems highly radioactive. It can cause problems for the final product (the proposed plaster) because, due to the high radioactivity, the amount of material made of PG in construction would be reduced, limiting its use if the Canadian standards are respected (150 Bq/m3: a level that it is recommended not to exceed) [27]. However, the radioactivity of Tunisian PG is lower than the average level of PG in the world, mainly due to the presence of radium 226 (209 $\pm$ 8 to 223 $\pm$ 8 Becquerels/kg), which is lower than that of Morocco (1032 Becquerels/kg), Serbia (488 $\pm$ 15 to 737 $\pm$ 8 Becquerels/kg), and others (Table 3). The weak secondary radioactivity is due to the insertion of uranium-238 and thorium-232 into the acidic portion. This result is consistent with Ajam et al. [28]. Thus, this leads to lower costs for processing Tunisian PG treatment compared to other countries. Furthermore, by using only 10% per volume of PG, we can achieve the highest fire resistance of GPS plasters and build a larger area of the walls. A proportion of more than 10% PG in the manufacture of building materials leads to increased radioactivity concentration in the building (which is, of course, undesirable).

**Table 3.** Radioactivity of some phosphogypsum in the world (Becquerels/kg).

| Origin of PG | U$_{238}$ | Ra$_{226}$ | Th$_{232}$ | References |
|---|---|---|---|---|
| Tunisia (Gabès region) | 45.4 | | 18.9 | [9] |
| Tunisia | 35 $\pm$ 2 to 66 $\pm$ 4 | 209 $\pm$ 8 to 223 $\pm$ 8 | 8 $\pm$ 1 to 20 $\pm$ 2 | [29] |
| Morocco | 1191 | 1032 | Below detection limit | [30] |
| Morocco (use in Spain) | 140 | 620 | | [31] |
| Brazil | 17 $\pm$ 5 to 42 $\pm$ 6 | 100 $\pm$ 7 to 695 $\pm$ 47 | 156 $\pm$ 38 to 175 $\pm$ 23 | [32] |
| Flodida, USA | 130 | 1120 | 3.7 | [33] |
| Egypt | 134 | 411 | 19 | [34] |
| Serbia | - | 488 $\pm$ 15 to 737 $\pm$ 8 | 2.1 $\pm$ 0.8 to 4.5 $\pm$ 0.9 | [35] |
| Limit values | $1.10^4$ | | $1.10^3$ | [36] |

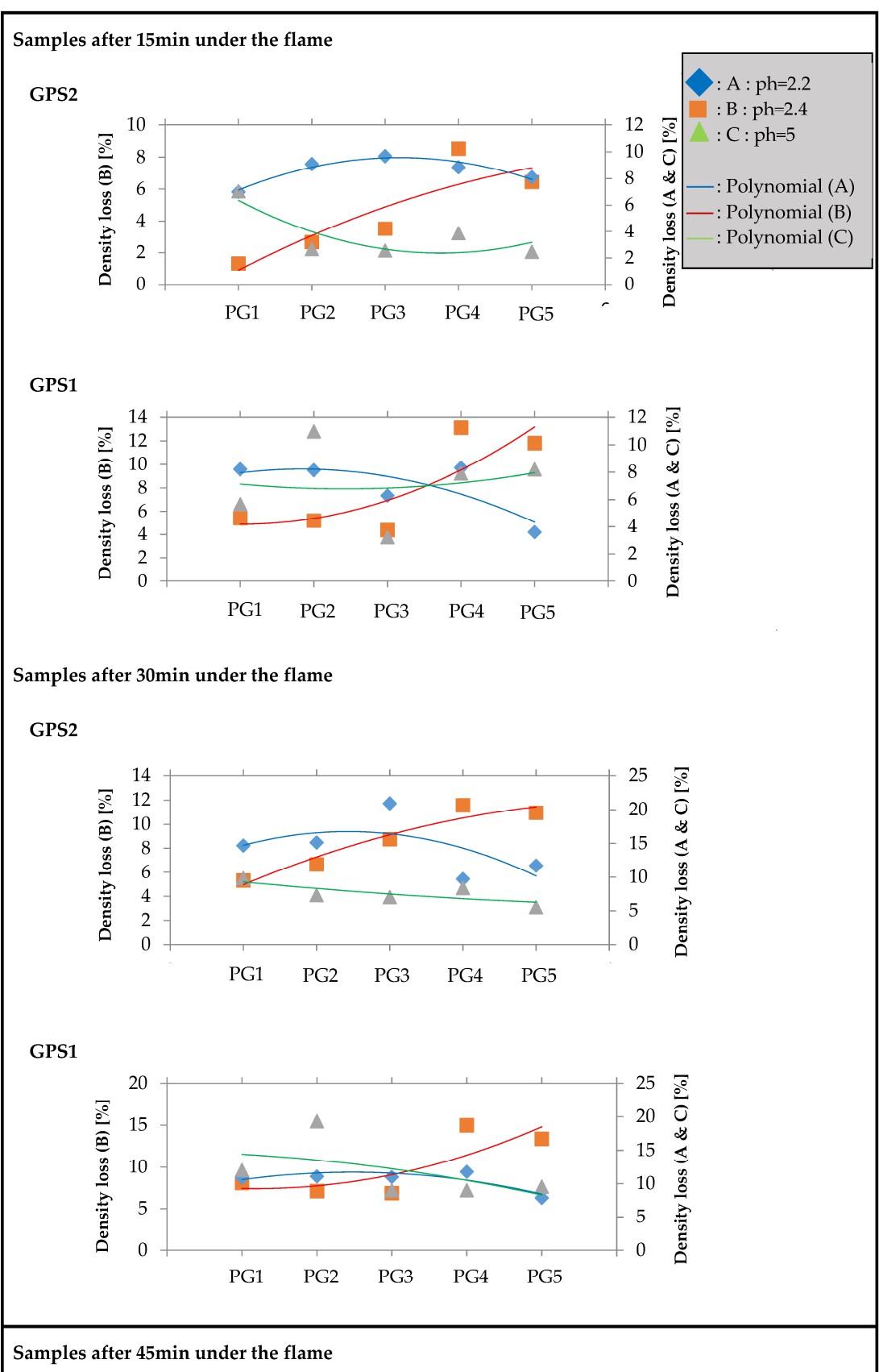

**Figure 7.** *Cont.*

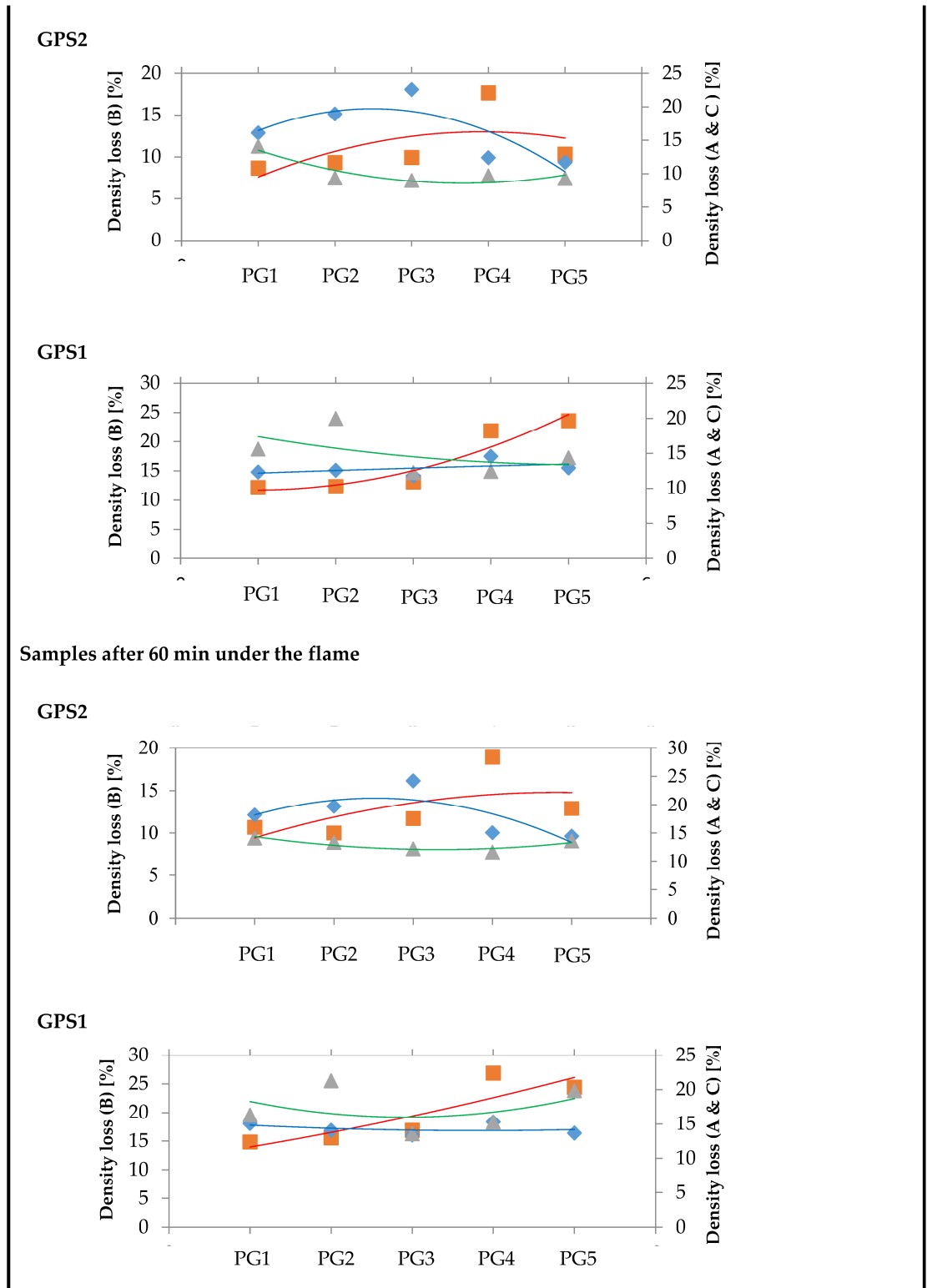

**Figure 7.** Variation in density loss of plaster under the flame in different periods and plans.

On the other hand, we must consider protecting drinking water resources, avoiding their use in washing processes, and eliminating radioactive water from PG washing. For neutrality, we used distilled water to prevent the influence of the composition of impure water (drinking water, seawater, etc.) on the results because researchers have shown that washing PG twice with drinking water increased the Mg content from 571% to 1294% com-

pared with untreated water [26]. We donot recommend using drinking water to purify PG in practice as it can exacerbate the drinking water crisis in countries. While the use of seawater treated, even if relative, can be an effective solution, chemical analyses must be carried out each time because the chemical compounds of the washed water change as the wash water changes. They will not be expensive (since they will be used to clean PG, not drink), especially since PG is very acidic (it becomes an alkaline pH material). At the same time, PG helps reduce seawater's salinity by up to 19%, which could result in positive effects on aquatic life [37]. After cleaning the PG, it is not recommended to dispose of this leachate or to reuse it without treatment, as cleaning the PG will lead to the leaching of radioactive elements. The researchers could use organic or inorganic materials with a high absorption capacity for these unwanted elements to reduce radiation and treat water pollutants extracted from factories. For example, oxidized biochar fibers from cylindrical loofah sponges exhibited high adsorption capacity for TH (IV) ($Q_{max} = 70$ mg$^{-1}$) [38]. Dueto these measures, we can at least help reduce the proportion of radioactive contamination and keep drinking water.

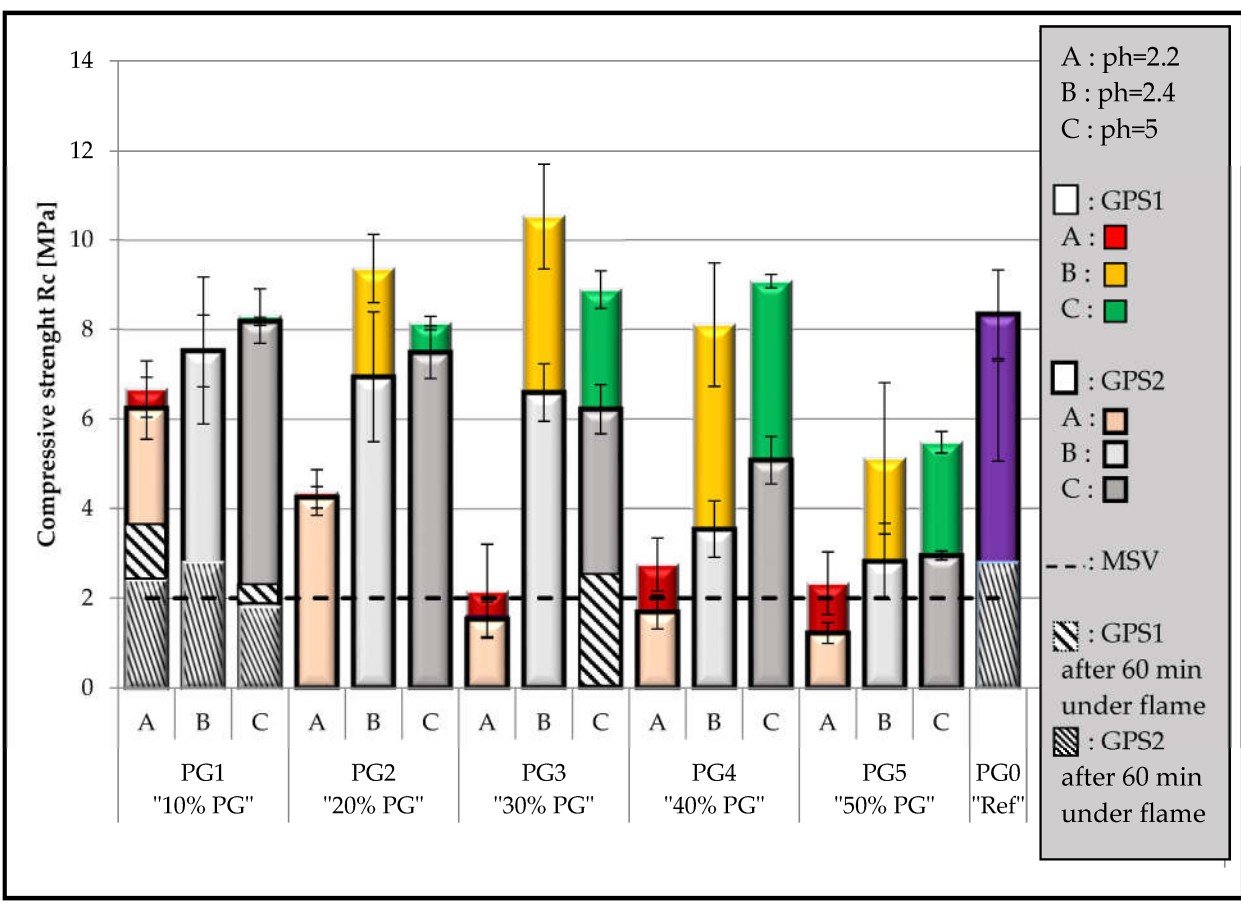

**Figure 8.** Variation in compressive strength of plaster in the different plans, before/after 60 min under flame.

## 3. Materials and Methods

### 3.1. Materials

3.1.1. Phosphogypsum Pre-Treatments

We prepared samples according to British Standard EN 13279-1 using only two materials, GPS (GPS1 or GPS2), as received PG or washed PG [39]. The plaster used is a commercial product extracted from southern Tunisia, one from the "Mestaoua" geological formation in the Tataouine region (GPS1) and another from the Sidi Bouzid region (GPS2).

GPS deposits in Southern Tunisia are known for their high availability and purity, especially Mestaoua GPS. Table 2 characterizes plaster from Southern Tunisia [8].

PG is "unnatural plaster". GCT provides it from the Gabès region in Tunisia. PGs are less than 250 μm in diameter. The PG powder was dried at 100 ± 5 °C for 24 and 48 h, and the raw and washed PG yielded constant moisture contents of 36% and 54%, respectively. The bulk densities of natural and washed PG were 0.80 and 0.86 g.cm$^{-3}$, respectively. The chemical formula of PG is as follows:

PG is washed (WS) several times with distilled water (WS0, WS1, . . . , WS5) to reduce its chemical impurities and partially eliminate its acidity (Figures 9 and 10). According to mixing schemes A, B, and C, only three kinds of PG solid residues are used in GPS production: raw PG (pH = 2.2), PG after WS2 (pH = 2.4), and PG after WS5 (pH = 5). In practice, we stirred PG in 4 L of distilled water for 1 h, using a magnetic stirrer for each wash (Figure 9), according to the PG treatment protocol described by Moncer et al. [26]. The washing process was repeated five times. The pH measurement is carried out by a pH meter at the end of each wash.

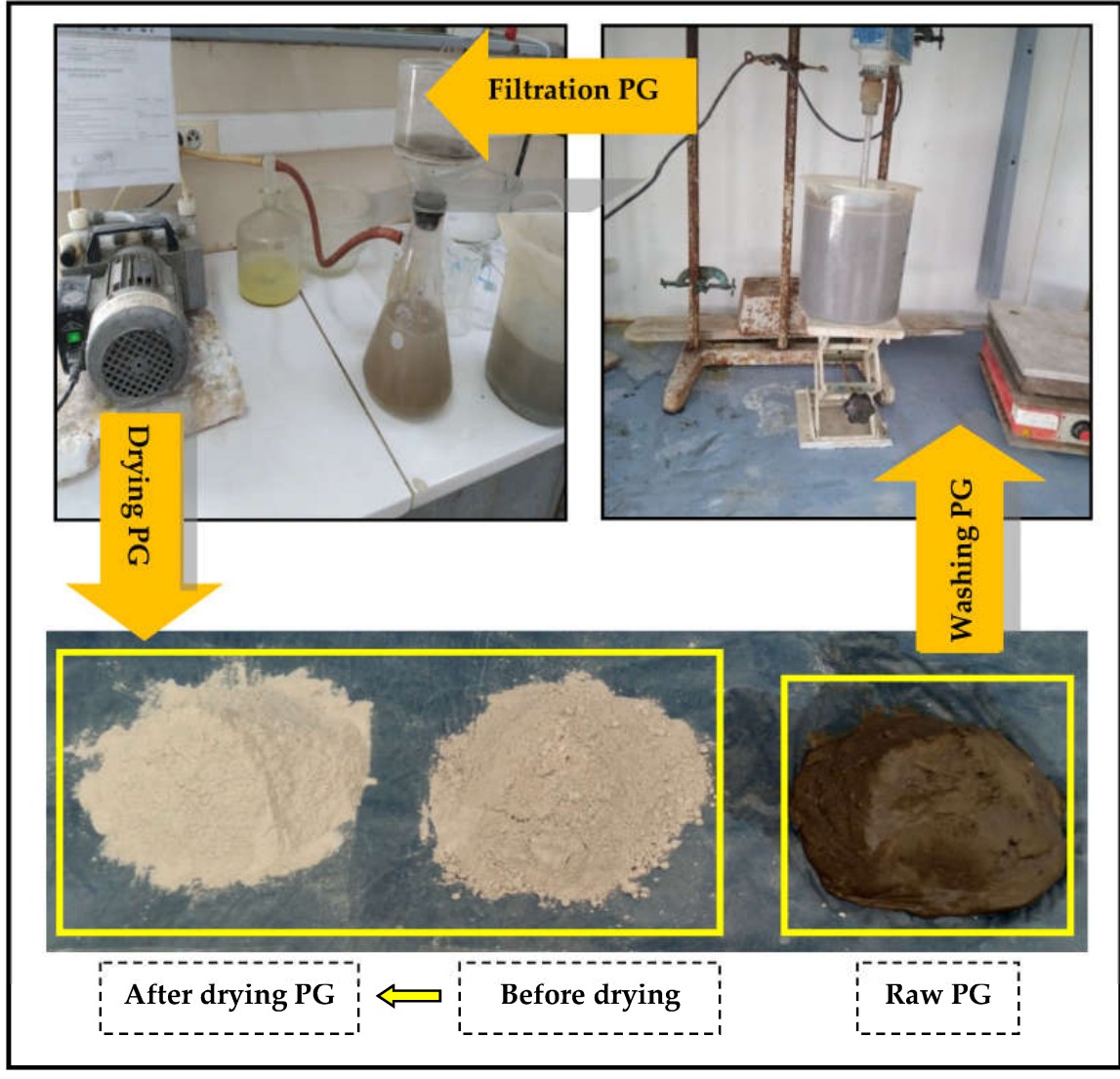

**Figure 9.** Phosphogypsum preparation cycle.

PG powder was recovered through a 100 μm diameter sieve after each wash. Collect PG particles smaller than 100 μm in diameter with a vacuum pump to reduce the loss of the amount of PG during washing (Figure 9). The raw and washed PG was dried at a

temperature of $100 \pm 5$ °C to obtain a dry powder that could be used to prepare plaster samples (Figure 9).

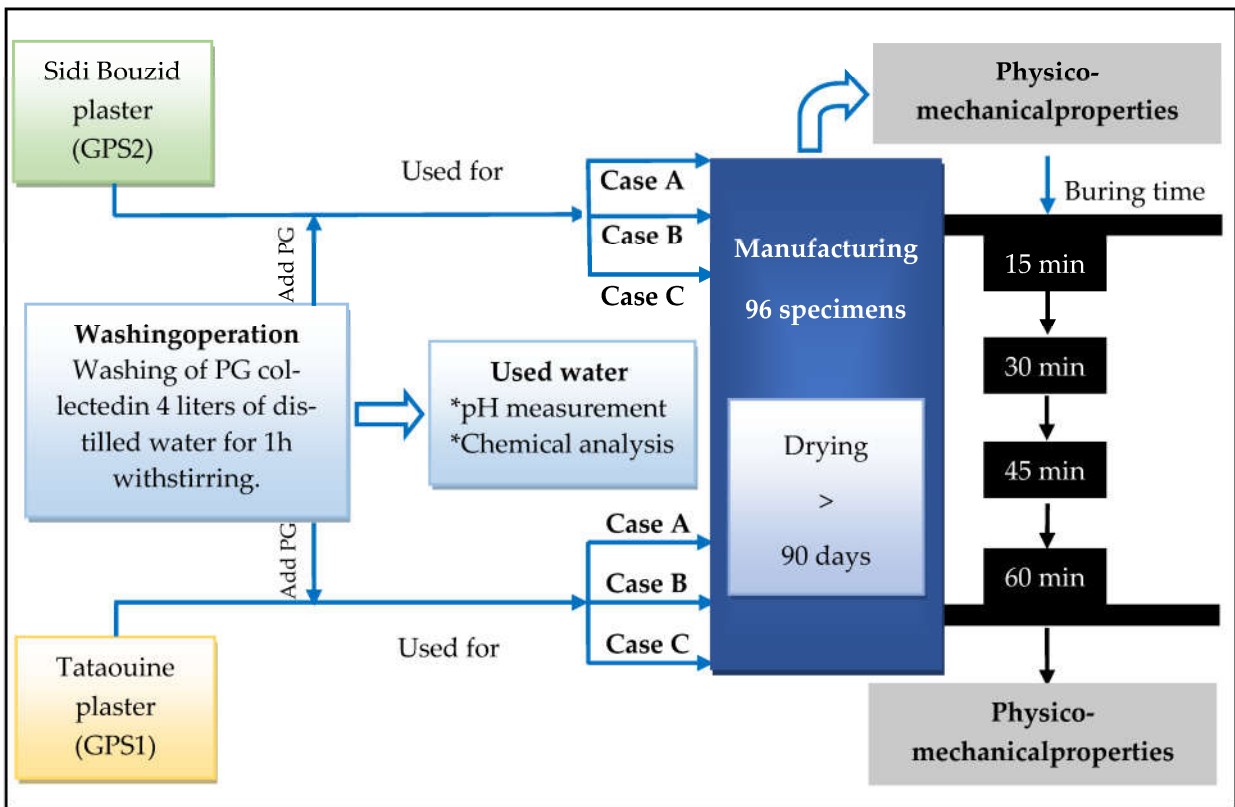

**Figure 10.** Clarification of the sequencing of this research work.

Making phosphoric acid from natural phosphates in factories produces radioactive PG because phosphates, like many minerals, are naturally radioactive. The researchers showed that Tunisian PG is less radioactive than other PGs in the world (Table 3). The activities of Thorium-232 (15 Bq/kg) and Uranium-238 (47 Bq/kg) of PG in Table 3 donot pose a risk to the environment [9]. These two elements (Uranium-238 and Thorium-232) were found to be preferentially retained in phosphoric acid, while the component (Radium-226) was found almost exclusively in PG [36]. The biggest environmental problem, Radium-226, is still stored in PG. Hamdi et al. confirmed that the measured radioactivity of raw PG did not exceed the limits set out in the EU Council Directive "2013/59/Euratom", which establishes basic safety standards for protection from ionizing radiation [9,40]. However, it is important to be aware of the misuse of PG in construction, as it is a radioactive substance, since the higher its proportion in the manufacture of the material, the higher the proportion of radiation, thus exceeding the limits of health protection.

3.1.2. Composite Preparation

GPS-based specimens (GPS1 and GPS2) were fabricated using the following different mixing plans (Figure 10):

- Case A: Mixture of plaster with raw PG (WS0, pH = 2.2);
- Case B: Mixture of plaster with washed PG (WS1, pH = 2.4);
- Case C: Mixture of plaster with washed PG (WS5, pH = 5).

In all mixing cases, for each sample, we performed the following steps:

1. First, mix the plaster and PG dry powder with a vacation step equal to 10% (Table 4);
2. Next, add mixed water to keep the water/plaster ratio constant (0.75) while mixing the mixture well for about 5 min;

3. Then, add the mixture into a standard-sized mold of $4 \times 4 \times 16$ cm$^3$;
4. Finally, after checking its hardened state, the specimen is removed from the mold on the day of casting.

**Table 4.** Various cases of phosphogypsum mixed with plaster.

| Samples | Cases | XmP * | XmPG * | W/P * |
|---------|-------|-------|--------|-------|
|         | Raw/Washed | [wt%] | [wt%] | |
| PG0 *   |       | 100   | 0      |       |
| PG1     | A<br>B<br>C | 90 | 10 |       |
| PG2     | A<br>B<br>C | 80 | 20 |       |
| PG3     | A<br>B<br>C | 70 | 30 | 0.75 |
| PG4     | A<br>B<br>C | 60 | 40 |       |
| PG5     | A<br>B<br>C | 50 | 50 |       |

* PG0: Reference sample. * W/P: The water/plaster ratio. * XmP: Weight percentage of plaster. * XmPG: Weight percentage of phosphogypsum.

According to Table 4, 96 samples were prepared. Allow these samples to dry for more than 90 days at room temperature in the laboratory.

*3.2. Characterization Methods*

3.2.1. Mineralogy and Morphological Characteristics

X-ray diffraction (XRD).The mineralogical composition of GPS was determined by XRD analysis on powder preparations with a Siemens D500 diffractometer (40 kV, 40 mA, CuK$\alpha$ radiation).

Scanning electron microscope (SEM). The morphology of GPS and plaster were examined with a Quanta 650 FEG scanning electron microscope.

Chemical analysis of phosphogypsum. The chemical analysesof elements ofPG; Mg, Cd, Fe, and Cawere carried out using an atomic absorption spectrometer (AnalytikjenanovAA 300).The phosphate, sulfate, and fluoride concentration were measured using ionic chromatography (930-compact-ic-flex).

3.2.2. Mechanical Properties Testing

Mechanical resistance. The calculations of the mechanical properties of the samples, both compression, and flexural resistance, were carried out according to NF EN 196-1 [41], such as:

Flexural strength formula:

$$R_f = \frac{1.5 * F_f * L}{b^3} \tag{2}$$

where: L: Distance between two supports equals 10 cm, $R_f$: Resistance to bending in MPa, $F_f$: Breaking load in N, b: Width of the square section equals 40 mm.

Compressive strength formula:

$$R_C = \frac{F_C}{b^2} \tag{3}$$

where: $R_c$: Compressive strength in MPa, $F_C$: Breaking load in N, b: Width of the square section equal to 40 mm.

Fire-resistance test. We tested the reaction to fire on 32 samples taken after a flexural strength test. The combustion of these samples was carried out with a gas flame under almost the same realization conditions (Figure 11):

- Heat flux (heat passing through a surface over time);
- Burn duration (15, 30, 45, and 60 min);
- The distance between the flame and the sample (about 30 cm);
- Sample thickness (40 mm).

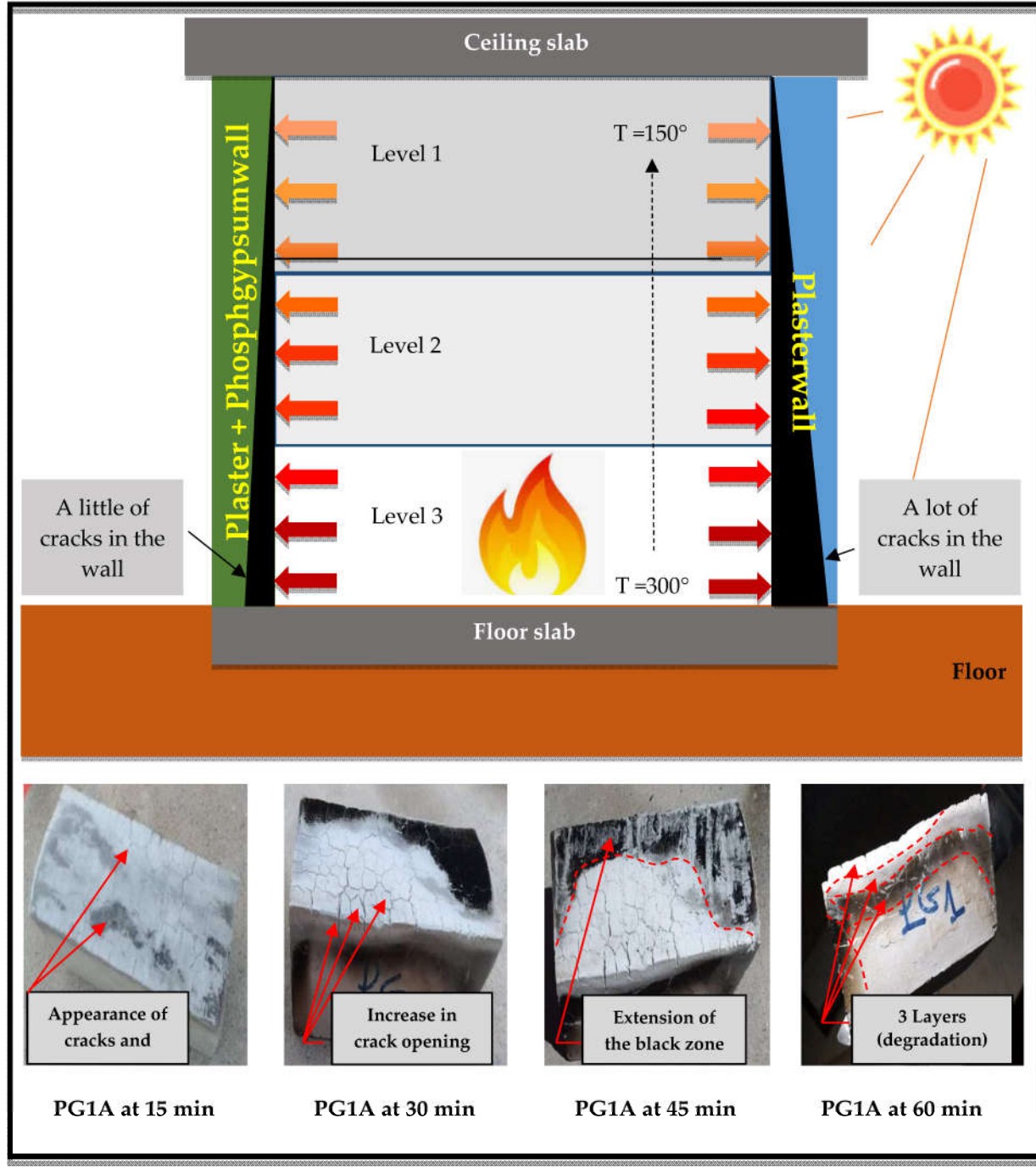

**Figure 11.** Fire-resistant test.

This demonstration is only intended to demonstrate the difference in fire performance between GPS 1 and GPS2 samples using raw and washed PG. However, ISO 11925-2 recommends using a flame when burning the material [42]. The fire resistance rating is carried out according to the following standards:

- The mass fraction lost (preservation of structure after each fire duration);
- The compressive strength of the samples which did not break after 60 min of burning.

Bulk density ($\rho_{app}$). The density and mass loss for each combustion duration in Figures 4, 5 and 7 are determined by the "weight/volume" ratio according to European standard NF EN 1097-6 [43].

## 4. Conclusions

In this work, we succeeded in developing a new eco-friendly composite. Contaminated materials can profit suitably based on simple ways to save lives before buildings collapse. It is to continuously wash phosphogypsum (PG), use dehydrated Tunisian gypsum (GPS), and mix these two materials in different proportions to obtain plaster with excellent mechanical properties and good fire resistance. The proposed application can be used in various fields (agriculture, construction, medical equipment, etc.), especially for the construction of interior walls of buildings, without adding other products, such as fibers, sand, product chemical, etc. In practice, our plasters provide reliable value. The practical results show that the mixture of 10% PG with GPS1 increased the compressive strength (7.34%) and the flexural strength (15.66%) compared to the reference sample. In terms of fire resistance, at the exact PG dosage, our GPS maintains the structure of the sample well (15.07% loss of mass at 60 min under flame) and does not exceed the minimum compressive strength allowed by the standard even when exposed to flame for 60 min (3.6 > 2 MPa).

**Author Contributions:** Conceptualization, A.G. and N.H.; data curation, A.G., F.A., N.M. and O.B.-U.; formal analysis, A.G., F.A., N.M. and O.B.-U.; funding acquisition, J.-F.W.; investigation, A.G., N.H., O.B.-U., J.-F.W. and M.F.Z.; methodology, A.G., J.-F.W. and N.H.; project administration, A.G., N.H. and J.-F.W.; supervision, M.F.Z.; validation, A.G., J.-F.W. and M.F.Z.; visualization, N.H.; writing—original draft, A.G.; writing—review andediting, N.H. and O.B.-U. All authors have read and agreed to the published version of the manuscript.

**Funding:** The publication was funded by the Open Access Fund of the University of Trier and the German Research Foundation (DFG) within the Open Access Publishing funding program.

**Data Availability Statement:** Not applicable.

**Acknowledgments:** The authors thank the Department of Civil Engineering at the Higher Institute of Applied Sciences and Technology of Gabès and the Tunisian Chemical Group for their support of this research project, as well as Yannick Hausener, Technician, Geology Laboratory, Department of Geology at the University of Trier for his mission. The research presented in this manuscript is part of the thesis of Abdessalam Guedri.

**Conflicts of Interest:** The authors declare no conflict of interest.

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
