# Peer review of "Addition of Phosphogypsum to Fire-Resistant Plaster Panels: A Physic–Mechanical Investigation"

_inorganics, doi:10.3390/inorganics11010035_

Round 1
Reviewer 1 Report
The manuscript presents the results of an investigation performed on the valorization of phosphogypsum (PG) in combination with regular plaster for the construction of fire resistant panels. The authors adopt the thermal treatment of the PG prior mixing to regular plaster till 50 wt%. The number of analytical techniques presented is more than sufficient to represent an original, novel and complete study. Nevertheless, the result presentation is really confusing and requires a complete reorganization. The manuscript should be re-written in most of its parts due to big problems in the use of the English language. Therefore, I suggest major revision and a complete reorganization of the manuscript.
My only perplexity is the adoption of a washing procedure that might produce leaching of radioactive elements. Comments of the leachate water is not discussed in the paper.
In the following, authors can find my detailed comments and suggestions.
TITLE
Title is too long and some words are redundant. What about: “Addition of phosphogypsum to fire resistant plaster panels: a physico-mechanical investigation”?
ABSTRACT
In the abstract, it is not necessary to indicate acronyms if they are not used in the text of the abstract. Hence, I suggest deleting the first (GPS) and leaving the remaining acronyms.
In the sentence: “The PGs were then washed several times according to the protocol” author either specify the protocol or delete the reference to it according to the protocol .
There are typing errors for ph, it should be: pH.
There are also typing errors for pH values, is it 2.2 or 2.4 ?. Please identify only one pH value to refer to.
INTRODUCTION
Authors should identify the proper nature of Phosphogypsum, if it is a secondary raw material or it is a waste. They should use this definition throughout the Introduction Section. The definition “contaminant” actually used by the authors is not appropriate.
Equation (2) should nicely fit after line 51. Moving the reaction in this part of the Introduction would help the reader to better understand the chemical background of the argumentation.
Line 52: The sentence: “The global PG release in 2018 was estimated at 230 tonnes.” deserve a reference to support this data.
Line 56: “Phosphate surplus….” Do you mean Phosphate production?
Line 57: “More than 80% of phosphate production in Tunisia is converted into commercial products (acid (H3PO4, H2SO4, ect.),.....”. Are you sure to mention H2SO4??????
Line 64: “In the Gabès region of Tunisia, GCT industrial waste exceeds 150 MT of PG in the Mediterranean Sea.” Do you mean that the company GCT pour PG in the Mediterranean Sea?
Line 84-85: Please add a reference to this statement.
Line 100: the refractory board mentioned in this sentence if made from plaster? Please specify.
Line 102: “In fact there are some interesting unknown facts about PGs……”, do you mean that: “there are aspects which are not yet being investigated”?
Lines 102-105: this sentence is too long. It must be divided into sentences and the objectives of the investigation should be better explained.
There are several typing errors to correct and the use of the English language should be improved. Here are some suggestions:
Lines 32-33: Gypsum is a natural calcium sulfate hydrate (GPS, CaSO4.2H2O), which is converted to calcium sulfate hemihydrate (P, CaSO4.1/2H2O) when heated. The hemihydrate form is known for its low bulk density and fairly good ductility.
Line 43: The plaster is used in various fields such as construction..........
Ond many others....with the support of an English speaking person, authors could improve the quality of the manuscript enormously.
Figure 1 should be moved in the second page of the paper, since it is described very early in the text.
2. MATERIALS
This paragraph contains the description of the starting materials (precursors) used to prepare the sample (final plaster mixes).
Please, divide this Section in sub-sections:
2.1 PG pre-treatments
Lines 111-141 plus lines 160-174 (Radioactivity).
2.2 Composites preparation
Lines 143-159.
Additional comments:
Line 111: “....according to British Standard EN EN 1”, Are you sure to have mentioned the standard correctly?
Line 112: In this Section I suggest to use the definition: “as-received PG“ instead of “raw PG”.
Lines 112-114: “The plaster used is a commercial product produced in the Tataouine region (Southeast of Tunisia, the Mestaoua geological formation) and Sidi Bouzid region (southern central Tunisia).” Is the plaster used in this work just one? Why do you mention two different areas? Please be more precise in sample provenance.
Line 116-117: Please delete reaction (1), which has been explained in the text (line 32-33) and it is trivial.
Line 123: Complete Table 1 indicating the testing method used to measure each different property of the plaster. In Table 1: “Refusal at 200 micrometers” is intended to be: RESIDUE at 200 micrometers?
Line 124: Please provide town and country for GCT. Substitute the word “microns” with its symbol.
Line 128: Move equation (2) to the Introduction.
PLEASE MOVE HERE Lines 160-172, Table 3 included, to complete the description of the as-received PG.
Line 161: “PG is obtained from the production of phosphoric acid, so it is the production of natural phosphate.” This sentence is not clear.
Lines 167-169 : “In fact, the biggest environmental problem, Radium-226, is still stored in PG. Hamdi et al. confirmed that the measured radioactivity of raw PG did not exceed the limits set out in the EU Council Directive "2013/59/Euratom",”. It is not clear if the PG that you have used in this study is radioactive or not. Please clarify this point.
After completing the description of the PG, you can proceed with the description of its washing procedure.
Line 129: “.....to dissolve its salt chemistry….”, this part is not clear. Do you mean to “dissolve the soluble salts”???
LIne 130: Figure 4 (top part) is not clear. Add info in the text about WS1, WS”, …till WS5, and delete the top part of Figure 4.
Line 157: Table 2 will be renumbered as Table 3. Indicate is % is wt% in the brackets. There is no need to add *. Please describe in Table’s caption the acronyms: XmP and so on.
Figure 4: please keep only the bottom part and add details, as in the text.
5. Experimental methods
This Section should be renumbered as: 3. Characterization methods (lines 173-224).
Move Bulk density after Fire-resistant tests.
Check the definition of the standard procedures in the text (they do not correspond to those reported in the Refs).
Use min as an abbreviation for minutes. Also in Fig. 3.
6. Results and discussions (Please renumber this section.)
X-ray diffraction (XRD). Provide chemical formulas of crystalline forms individuated.
Chemical components. Table 4: Indicate the analytical procedures used to detect the amounts of cations and anions in the Table caption. Adopt the . in place of , in the numbers reported in this table.
Acidity and weight loss. Please separate the results of acidity and weight loss, when possible. Figure 7: Please add error bars.
Reviewer 2 Report
General comments:
A deep discussion should be added to this work, with its current version, it barely has results and nothing more. The abbreviation should be carefully checked and should be given the first time it appears. Typo mistakes are also presented in this manuscript, please correct them. I would suggest the authors merge the Materials, Composite preparation, and Experimental methods into a section as Materials and methods. And please consider moving the context related to radioactivity since none of the data is from your experiments. Meanwhile, in the introduction, I suggest adding more content assigned to fire resistance and mechanical properties variation of previous PG applications instead of using much space for introducing its inherent impurities.
Specific comments:
1. Abstract: Since you used ‘GPS’ to represent gypsum at the beginning of your abstract, please unify it in your following statement.
2. Abstract: Please extrapolate your main findings and outputs to a broader community that would be interested in this work.
3. Abstract: Line 18, is it volume fraction or weight percentage?
4. Lines 45-46: Please cite references since you used ‘current scientific trends’ here.
4. Line 52: which ‘is’
5. Line 58: ‘etc.’
6. Line 123: Why did you cite a publication in your results? Is the whole table from a published work?
7. Line 159: Please specify the ‘laboratory conditions’
8. Line 160-172: I cannot entirely agree that you used Tunisia phosphogypsum data to represent the radioactivity of your collected phosphogypsum.
9. Line 242: How did you detect and quantify the elements' concentrations? This information should be provided in Material and methods.
10. Line 285: Based on the error bars given in Figure 8, I don’t think the variation in density is so pronounced. In addition, please explain the meaning of columns with different colors in your figure caption.
11. Line 326: Please check the figure caption. (the orange square and the red square). Besides, I would recommend that you carry out a principle component analysis of the data provided in this figure. In the current version, it is very hard to read.
12. Line 359-361: I would suggest not using ‘more safer, simpler, and cheaper’. Because you didn’t carry out any TCLP experiments or radioactive elements characterization, so it is hard to say it is safe. And concerns ‘cheaper’, at least a short cost analysis is required prior to this statement.
Reviewer 3 Report
- The topic is extremely interesting and particularly in the EU there seems to be a strong drive to use phosphogypsum to replace natural gypsum (https://doi.org/10.1016/j.resconrec.2022.106328)
- I do not agree with the statement that PG is of equal value than natural gypsum. PG in the US for instance is treated as a waste product, while natural gypsum is not. I am sure you are aware of all this. Just make sure the statement is a bit more nuanced…
- Line 19: just washed with water I assume? Please specify
- Line 36: what is the source of these production numbers, is this per year? Please provide references…
- Line 53: again references please
- Line 56: 2010 is too old, provide newer data, and references…
- I would also mention that PG is low-radioactive…perfect, I see you do this in table 3. Maybe add more data here. For Morocco there should also good data available…Australia is not a major phosphate rock producer, I think it would be good if you could cover the major ones…
- The radioactivity of the PG from Tunisia seems very high, is this a problem for your final product? Please elaborate.
-
Round 2
Reviewer 1 Report
After reading the reply of the authors and the final version of the manuscript, I appreciated the work done the authors, yet I do have ethical concerns about this study.
The use of a washing procedure that consumes clear water and contaminates it with radioactive elements seems anti-ecological.
Additionally, the sentence:
"In the Gabès region of Tunisia, the company GCT pour more than 150 MT of PG in the Mediterranean Sea."
still represent a point of great concern. I'm afraid that this particular sentence might raise some questions from some environmentalists since it reports an improper action from part of a company,. By the way, this sentence is not necessary to clarify the scientific motivation of the authors, thus can be deleted without subtracting any real scientific info to the manuscript.
Writing a scientific paper is not the proper way to report improper behaviours.
Reviewer 2 Report
The authors have made careful revisions according to the comments of all reviewers. But I have very few suggestions as follows:
Line137 ‘μm’ in diameter?
Line 185 ‘pourcentage’?
Line 257 I would suggest using the formula ‘CaSO4· 1/2H2O’ to represent bassanite
Line 260-264 Please check the chemical formula of your ion pair
Line 402-403 Which chemical element, please specify
Reviewer 3 Report
- Thank you for addressing the reviewers comments. I can see that you put a lot of work in this. Please find below some more (minor) comments to further improve this work:
- Define GPS when it first appears in the abstract
- Why is Tunisia not producing more GPS? – one quick sentence is enough
- Line 86: I think this is “recovery” not elimination of lanthanides
- “Egypt” not Egypte
- The reference in Table 3 for Morocco is pretty old, published in 2001. You can add more up to date reference from Morocco: U238=1191 Bq/kg, Ra226=1032 Bq/kg, Th232 (below detection limit): Arhouni, F.E., Hakkar, M., Mahrou, A. et al. Better filterability and reduced radioactivity of phosphogypsum during phosphoric acid production in Morocco using a fly ash waste and pure silica additive. J Radioanal Nucl Chem 331, 1609–1617 (2022). https://doi.org/10.1007/s10967-022-08235-y
